# Is There an Association between Being a Victim of Physical Violence by Intimate Partner and Binge Drinking in Men and Women? Secondary Analysis of a National Study, Peru 2020

**DOI:** 10.3390/ijerph192114403

**Published:** 2022-11-03

**Authors:** Francesca Campoverde, Micaela de las Casas, Dora Blitchtein-Winicki

**Affiliations:** Health Science Faculty, School of Medicine, Universidad Peruana de Ciencias Aplicadas, Av. Alameda San Marcos 11, Chorrillos 15067, Peru

**Keywords:** binge drinking, physical aggression victimization, intimate partner violence, male, female, Peru, demographic health survey

## Abstract

The relationship between being a victim of physical violence by an intimate partner and binge drinking (BD) is a poorly explored line of research, especially in men. To determine the association between being a victim of physical violence by an intimate partner and BD in men and women in Peru in 2020, a secondary analytical cross-sectional study was conducted using the Demographic Health Survey. BD was categorized according to the Center of Disease Control definition for men and women, based on the type and amount of alcoholic beverage ingested. Physical violence was based the report of being hit with any part of the body or an object, by their intimate partner. To identify the association, a multivariable general linear model of the family and link log Poisson was used. The results were presented as prevalence ratios (PRs). In the adjusted models stratified by sex, a 90% greater likelihood of BD was found in male victims and an 80% higher probability among female victims (PRa 1.9, 95%CI 1.3;2.7, *p* < 0.001 vs. PRa 1.8, CI95% 1.1; 2.8, *p* = 0.013, respectively). An association was found between physical violence by an intimate partner and BD in the Peruvian population older than 15 years, in both men and women.

## 1. Introduction

Intimate partner violence is a normalized public health issue in many cultures worldwide, especially in developing countries [1,2,3,4]. According to the World Health Organization, one in three women in the world has suffered physical and/or sexual violence at some point in her life [5]. However, little is known about the actual proportion of male victims of intimate partner violence, especially in developing countries [6,7,8,9]. There is an information gap regarding its global prevalence [10], although 30% of men have reported suffering it in the United States [10]. In Peru, information on male victims of intimate partner violence is not as widely studied as in women, neither does include psychological or sexual violence [4]. In 2019, reports showed that 10.7% of women had been victims of physical violence by their intimate partners during the last 12 months [11]. Among the sequelae of violence mainly identified among women are physical injuries, as well as traumatic experiences, some of them permanent. Regarding relationships and interaction with society, victims of intimate partner violence are more likely to have depression and other mental health disorders [12,13], engage in risk behaviors, and contract sexually transmitted diseases [14]. They are also more likely to experience a reduction in their quality of life. In addition to these direct effects on the victims of physical violence, there is a great economic burden on the health system and on productive activity [15]. Despite the above, it is regarded as an everyday issue, and due to the stigma associated with being a victim, the different gender roles in each culture, and the lack of support networks for victims, the latter often choose to remain silent, which contributes to the chronification of the invisibility of this problem, mainly in developing countries [16].

A longitudinal study conducted on young people in Peru during the mandatory lockdown due to the COVID-19 pandemic in 2020, identified that 8.3% of this population experienced an increase in domestic physical violence without significant differences regarding sex, which was more significant in those individuals with a history of violence [17]. Social isolation and quarantine are associated with an increase in anxiety and stress and a drastic change in routines and habits, such as increased alcohol and tobacco consumption; these being risk factors that contribute to the increase in aggressive behaviors [18,19]. A study analyzing the characteristics of the COVID-19 pandemic and the lockdown context reveals that the increase in stress and maladaptive coping strategies, such as substance abuse, may lead to aggression and an increase in mental health disorders, which have been directly related to the escalation of intimate partner violence [20]. A direct relationship between global stress during the COVID-19 lockdown period and intimate partner violence has also been identified [21]. A systematic review investigating the changes in the prevalence and severity of domestic violence before the pandemic, and during the social mobility restrictions in 2020, found that, despite the increase in the severity of emotional, psychological, and sexual violence in the general population, no conclusive evidence could be found regarding changes in the prevalence of physical violence. This review also highlighted the scarcity of studies reporting men as victims [22].

On the other hand, alcohol consumption has been identified as one of the main risk factors for cancer, accidents, and communicable diseases, leading to premature death and disability worldwide [23]. The burden of disease related to alcohol consumption varies depending on the type of consumption: the highest average volume of alcohol consumed is more related to chronic consequences such as cancer, diseases derived from alcohol abuse, and liver and infectious diseases, while excessive consumption of alcoholic beverages in a short period, or binge drinking (BD), is more associated with ischemic and cardiovascular diseases, accidents, and communicable diseases [24]. BD varies in different populations and even between regions and is conditioned by social situations and norms, cultural, religious, and socioeconomic features; differences can even be observed throughout the life cycle. It is more common in men than in women and in early life stages (for example, during adolescence and youth) [25,26,27]. A relevant factor affecting BD is the alcohol content and the price of the different types of alcoholic beverages available in each area, which includes non-registered alcoholic beverages, such as fermented beverages and craft liqueurs [28]. BD refers to men consuming five or more alcoholic drinks and women consuming four or more within a two-hour period [29]. Increasing age, transition to different stages of life, greater accessibility, having home drinking habits, and tobacco consumption, as well as a low educational level, are all factors associated with BD [27,30]. Alcohol reduces the concentrations and sensitivity of brain serotonin and norepinephrine, depresses the nervous system [31], is detrimental to cognitive functioning, and causes disinhibition, which is why it is associated with the use of psychoactive drugs, aggressive behavior [28,30], drunk driving, and non-consensual sexual relations. Therefore, BD represents a public health issue worldwide [32].

As evidenced at a global level, intimate partner physical violence requires urgent attention and is associated with alcohol consumption [33]. On the other hand, being a victim of intimate partner violence is a traumatic event that leaves multiple chronic problems [15]. In Brazil, the male victim population showed excessive alcohol consumption [34]. Similarly, in Missouri, in 2015, intimate partner violence was found to be associated with smoking, a sedentary lifestyle, obesity, and BD in victims, as well as with a lack of motivation, fear, and trauma [35]. Among female victims from South Africa, excessive alcohol consumption was observed in 11.6% of cases, which was associated with childhood trauma, PTSD, depression, and bidirectional intimate partner violence [36]. Evidence suggests a bidirectionality and cyclical temporal dynamic between BD and IPV. The mechanisms in BD to victimization include settings where victims and perpetrators drink heavily, which increases disinhibited behaviors, as well as increased vulnerability to victimization. Contradictory IPV victims who commit BD after the aggression seek for an avoidant coping strategy, a distraction of psychological symptoms. Research in this last direction is less explored and there are some discrepant results, especially as these studies are been conducted predominantly in women [37].

In Peru, in 2018, 6.2% of women reported that their partners drank alcohol occasionally, while 52% of them stated that they had suffered physical violence when their partners were under the influence of alcohol [4]. Unfortunately, there are no values recorded for the male population. Although BD is known to exacerbate violent behaviors [32,35,36,38] worldwide, this reported consumption focuses on male perpetrators and female victims, but not on both female and male victims of violence in the same population [10,22]. For this reason, this study aimed to identify the association between physical violence by an intimate partner and BD in both male and female victims in Peru, during 2020.

## 2. Materials and Methods

This was a secondary analytical cross-sectional study that uses the ENDES (Demographic and Family Health Survey) 2020 database from the INEI (National Institute of Statistics and Information Technology). This study opted for a probability, balanced, stratified, and independent two-stage sample, which was carried out at the department level, considering urban and rural areas, and that regarded the dwelling as the sample unit [39].

The population consisted of Peruvian men and women aged 15 years or older in 2020. The study comprised those people meeting the inclusion criteria, such as being 15 years or older, being selected for the ENDES Health Questionnaire and having answered “yes” to the question of whether they had been married or in communal living situation in the last 12 months [40]. Those who did not answer questions regarding alcohol consumption and the frequency with which they had suffered physical violence by their intimate partners, with any part of the body or object during the last 12 months, were excluded. Lastly, pregnant women were also excluded [41,42].

### 2.1. Characteristics of the Primary Study

The primary study collected data about demographic health indicators in Peru [43]. The sampling framework of the ENDES 2020 was based on the statistical and cartographic information of the 11th Population Census and 6th Housing Census 2007, with the respective updates of the Household Targeting System (SISFOH, by its Spanish acronym) in 2012 and 2013, in addition to the 12th National Population Census and the 7th National Housing Census 2017 (PHC 2017) [39]. In this study, a probability, balanced, stratified, and independent two-stage sample was chosen. The study was conducted at the department level and based on urban and rural areas. Regarding urban areas, the sampling unit used considered conglomerate and private housing, while in rural areas, sampling was carried out based on the rural census area and private housing. For ENDES 2020, 37 to 390 homes were used: 15,098 dwellings in department capitals and in the 43 districts of metropolitan Lima; 9490 dwellings corresponding to the rest of the urban area and 12,802 dwellings from rural areas [39].

The power of the study was estimated using the statistical software Open Epi version 3.01, considering a 95% confidence interval and 14.5% prevalence of BD in female victims of intimate partner violence, and 36.3% of excessive alcohol consumption in male victims of intimate partner violence [43]. In the general population, 8.4% of women and 22.8% of men reported excessive alcohol consumption. Considering the information from ENDES 2020, which shows 238 men exposed to physical aggression by their intimate partner and 9685 unexposed, as well as 529 women exposed and 10,164 unexposed, the estimation was made by sex, and a power greater than 80% was identified considering a design effect of two.

### 2.2. Variables

The main independent variable was physical violence by the intimate partner in the last 12 months, according to sex. It was elaborated from questions asking whether the participant had been attacked by his or her partner in the last 12 months with any part of the body or an object, and the number of times. Each question was categorized as a dichotomous variable depending on the presence or absence of physical aggression by the intimate partner, considering “0” or “no” if the answer was never/do not remember, while “1” or “yes” was introduced when the answer was “rarely”, “sometimes” or “often”. Should the answer be “no” for both questions, the person was disregarded as a victim of physical violence by the intimate partner. Conversely, “yes” was recorded when any of the questions had an affirmative answer [40].

The dependent variable was excessive alcohol consumption in the last 30 days. The standard by the Center for Disease Control and Prevention (CDC) was used to measure it, which implies that a blood alcohol concentration between three and four grams increases the risk of suffering health consequences such as injuries, violence, accidents, and risky sexual behavior, among others [44]. A drink is defined as a beverage containing between 12 to 14 g of ethanol and that increases blood alcohol content by more than 0.08 g/dL [45].

In this study, the ethanol content was considered based on the type of alcoholic beverage consumed, to identify the number of drinks. One drink is equivalent to 300 mL of beer at 5%, 140 mL of wine or spirituous drinks at 12%, or 40 mL of liquors or distillates at 40% [46]. These data come from the ENDES question: “Have you drunk alcohol ‘more than once’ in the last 30 days? ”, and from the questions: “Think about the time you drank the most”, together with the specification of the type of alcoholic beverage, the quantity and number of glasses or bottles [40]. Excessive consumption was identified as “yes” or “1” if women had consumed the equivalent of four drinks or more according to the type and quantity of beverage on one single occasion during the last 30 days, or to the equivalent of five or more drinks according to the type and amount of drink in the case of men [45]. If the number of drinks was less, “no” or “0” was recorded.

Both variables considered a period of time, so that the physical violence by an intimate partner considers the last 12 months, while the excessive consumption of alcohol was measured by reporting the time that the largest amount of alcohol was consumed in the last 30 days.

Other variables considered in this study were the following: age in years; sex; educational level attained; marital status; area of residence (urban or rural); socioeconomic status classified in quintiles according to the characteristics of the dwelling; availability of specific goods and services according to the methodology adopted by the INEI [47]; ethnic self-identification; tobacco consumption in the last 30 days; and symptoms of depression in the last 14 days, measured by nine questions adapted from the PHQ9 scale with a score between 0 and 27, (0 to 4 being none or minimal symptoms, 5 to 9, mild symptoms, 10 to 14, moderate symptoms, 15 to 19, moderately severe symptoms and 20 to 27, severe symptoms). As a cut-off point, a score ≥10 was considered due to its high sensitivity and specificity for major depression [48,49], in addition to the type of alcoholic beverage most frequently consumed on more than one occasion during the last 30 days, such as beer, wine, anisette, etc.

### 2.3. Study Procedures

Procedures from ENDES to collect information on alcohol consumption and intimate partner violence included face-to-face interviews, which were conducted on site until March 16, due to the COVID-19 pandemic. Face-to-face activities resumed as of June 30. During this time span, the INEI adapted information from indicators that could be answered effectively by telephone. Likewise, during July to September, an information recovery strategy requiring attendance was implemented. Questions on intimate partner violence and alcohol consumption during the last 30 days, together with the type and quantity of drinks, remained in the face-to-face interviews [50].

The ENDES 2020 had a survey work team included a local supervisor, interviewers, and an anthropometrist [46]. The Health Questionnaire, which included the questions of the independent and dependent variables, considered one person aged 15 years or older per household, in which the ENDES was applied [46]. In the case of the dependent variable BD, the purpose was to know the amount of standardized alcohol drinks consumed in the last 30 days by people over 15 years of age, for which an Alcoholic Beverages Card was used to obtain the corresponding information. [46].

### 2.4. Statistical Analysis

The statistical analysis was performed using the STATA^®^ MP17 software with a confidence interval of 95%. For the entire analysis, the weights, primary sampling units, and strata were considered, using the svyset commands. For the descriptive analysis, the categorical variables were presented in frequencies, weighted percentages, and confidence intervals for the sociodemographic features and problematic behaviors in the population over 15 years of age in Peru in the year 2020. For the bivariate analysis, Pearson’s chi-squared test was used with Rao–Scott correction.

In order to identify the association between physical violence by an intimate partner and BD in the victim, a multivariable analysis was performed using a generalized linear model of the family, and Poisson log-link option in the crude and general adjusted model and in adjusted models stratified according to sex. Being a cross-sectional study, the results were presented as prevalence ratios (PRs). To enter the variables in the adjusted model, an epidemiological criterion was considered and multicollinearity was checked using the variance inflation factor with a reference value of 10. No higher value was found. Likewise, the correlation between the variables included in the adjusted model was reviewed using the estat vce command, correlation with a 0.5 criterion, and no correlation between the variables included in the adjusted models could be found.

The variables considered in the adjusted model were: age, socioeconomic level, sex, educational level, area of residence, ethnic self-identification, presence of depressive symptoms during the last 14 days, and tobacco consumption during the last 30 days [12,30,36,51].

## 3. Results

A total number of 20,616 people met the selection criteria for this study (Figure 1). Of the 11,317 participants who declared not having lived with a partner during the last 12 months, 6414 were women (54.2%) and 4903 were men (45.8%). As shown in Table 1, nearly seven out of 10 were 35 years old or older (66.8%), half of them were men (50.7%), almost five out of 10 identified themselves as being mestizos (48.4%), almost eight out of 10 lived in an urban area (78.9%), and more than two out of 10 had completed primary education or less (24%).

According to Table 2 and regarding problem behaviors according to sex, a higher percentage of women who had not consumed any type of alcoholic beverage in the last 30 days could be observed when compared to that of men (84.6% vs. 78.0%, respectively, *p* < 0.001). Women were found to present a greater depressive symptomatology (from moderate to more than moderate) than men (7.5% vs. 3.9%, respectively, *p* < 0.001). On the other hand, having been a victim of physical aggression by an intimate partner in the last 12 months was more common among women than in men (4.7% vs. 3.0%, respectively, *p* < 0.001).

Table 3 shows the association between sociodemographic characteristics and BD in individuals. The study found that 10.2% of the population BD. It is evident that the male sex (15.1%) has a higher rate of BD than female participants (7.1%). The population under 35 years of age presented higher rates of BD than those over 35 (30% and 17.8%, respectively, *p* < 0.001). Individuals who self-identified as Quechua and Aymara presented lower binge-drinking rates compared to Afro-descendants, Caucasians, and Mestizos (9.3% and 8.8% vs. 13.3%, 12.1%, and 11.7%, respectively, *p* < 0.001). People belonging to the bottom quintile were less prone to BD than those from the top quintile (6.3% vs. 14.4%, respectively, *p* < 0.001). On the other hand, coast inhabitants had a higher excessive consumption of alcohol compared to jungle and mountain dwellers (12.3%, 11.7%, and 8.0%, respectively, *p* < 0.001). In addition, people consuming tobacco in the last 30 days presented higher BD rates compared with those who had not consumed it (34.8% vs. 10.9%, respectively, *p* < 0.001).

With regards to the association between physical violence by an intimate partner and BD, Table 4 shows in the crude model that people who suffered physical violence by their intimate partner have a 90% greater likelihood of BD, when compared with those who did not suffer it (PRc 1.9, 95% CI 1.5–2.5, *p* < 0.001). Likewise, when adjusting for sex, age, self-identification, socioeconomic status, educational level, area of residence, tobacco consumption in the last 30 days, and depressive symptoms, the study found a 90% greater probability of BD (PRa 1.9, 95% CI 1.4–2.5, *p* < 0.001). In the sex-adjusted models, a 90% greater probability of BD was observed in men who were victims of violence, while in the female population the percentage was 80% (PRa 1.9, 95% CI 1.3–2.7, *p* < 0.05 vs. PRa 1.8, CI 1.1–2.8, *p* < 0.05, respectively).

## 4. Discussion

This study has found an association between being a victim of physical violence by an intimate partner and BD in men and women over 15 years of age in Peru. This association of victimization and BD is different from what was identified in a previous study in Brazil, where it could only be demonstrated in male victims, but not in females (OR 2.37, 95% CI 1.2; 4.9 and OR 1.17, 95% CI 0.6; 2.5, respectively) [34]. On the other hand, research conducted in the United States showed similar results as this study, revealing an association between being a victim of physical violence by an intimate partner and BD in both men and women [52]. In a longitudinal study of adolescents of both sexes, this association was also identified. One of the explanations considers that the personality and characteristics of people who engage in BD make them more vulnerable to becoming victims of intimate partner violence and choosing violent partners. Moreover, the social background in which this consumption occurs makes them more prone to sexual harassment, and their behavior under the influence of alcohol makes them even more susceptible to becoming victims of physical aggression by their intimate partner [53]. It is important to consider the COVID-19 context, since the research is based on data obtained throughout 2020, and therefore, most of this information was obtained under confinement conditions in which the couples’ relationships suffered modifications [17,19]. Similarly to the findings in the US, high levels of stress in couples during COVID-19 is related to a high number of physical violence attacks by the intimate partner, associated with risk behaviors such as those presented in this study [21].

As evidenced by other longitudinal studies [34,53], the abuse is not unidirectional, nor is it an isolated episode, but it is part of a cycle of abuse that can begin with violence or BD [54]. The latter can be the response by the victim or the perpetrator to the abuse [30], or a trigger for the perpetrator. However, the most frequently observed pattern in the victim is anticipation and fear of abuse [54]. Besides, behaviors against physical aggression are of the dose-response type. In other words, they are cumulative; the greater the violence, the riskier the behavior. It is important to highlight the contrast in the consumption between the rural and urban population, as people who live in rural areas tend to present lower excessive consumption of alcohol associated with a weaker culture of consumption and a more complicated access to alcohol [55]. Therefore, this is another factor that is included in the cycle of abuse, in addition to educational level, socioeconomic status, tobacco consumption [51], the presence of depression [56], age, and sex [34].

Some proposed mechanisms for understanding BD as a consequence of partner or ex-partner victimization in men or women include the interaction of factors at different levels, such as individual characteristics and background. For example, genetics, personality, the level of self-esteem, adverse childhood events, and alcohol consumption may condition a greater predisposition to conscious or unconscious binge drinking in response to negative psychological symptoms resulting from victimization, either from the incident or from incident-related problems, according to the dose-response hypothesis [57,58]. In addition to allowing that in some cases this excessive consumption, which was given as an action to buffer the symptoms, instead of achieving the desired effect, results in more negative emotions generating a vicious cycle [59].

This interaction also encompasses factors at the relationship level, such as the type of partner (e.g., current partner, ex-partner, same sex partner), alcohol consumption (habitual use, frequency of BD, relation to the perpetration of violence), and characteristics of the place where the violence and excessive alcohol consumption occur, whether in private and alone or in bars, parties or social gatherings, or others [60]. Further, social factors and the cultural norms of each setting have a role here, such as social stigmatization and the resources available to identify, prevent and manage IPV and BD situations, which may vary in different populations and cultures [37]. Additionally, alcohol is not exclusive to binging; food and substance abuse are included as well [61,62].

The female victims of intimate partner violence (whether physical, sexual, or psychological) most likely present schizoid, schizotypal, “*borderline*”, paranoid personality traits; they also have low self-esteem, little attachment to their families and 40% of them even blame themselves for the abuse [63]. Likewise, the cycle of abuse in women is associated with the presence of child abuse, depression, as well as with post-traumatic stress disorder [36]. The culture of alcohol consumption is less prevalent among women [23]. Although women also show excessive consumption of alcohol, social and work isolation, and depressive symptoms, are more common [30,56]. In the case of the male sex, there is often a previous history of child abuse, violence normalization, and a culture of domestic violence. For men, experiencing intimate partner aggression directly contravenes the generalized stereotype of the “superiority” of men over women in Western and patriarchal cultures [64], and their victimization is taboo, being therefore under-reported and stigmatized in most cultures and countries [64,65]. Their personality is associated with lability and being complacent. The most common type of violence is psychological [64]. Regarding men’s response to excessive alcohol consumption, it is globally known that too much drinking is synonymous with “drinking as a man”, so culturally, men have a greater tendency to consume in excess [64,66]. Consequently, when faced with aggression by an intimate partner, one of the most frequent coping strategies is BD [66].

In 2019, it was found that 10% of women between 15 and 49 years old had been victims of physical violence by their intimate partner in the last 12 months. Conversely, for 2020, this study focused on the population over 15 years of age according to sex, and found that 4.7% of women and 3% of men had been victims of intimate partner violence. Regarding alcohol consumption, the national indicators of the ENDES for 2020 report prevalence values of disorder by alcohol abuse in both men (5.2% of the population) and women (0.8%) [50]. A contribution of our study is the information about BD in 2020, which was 10.2% (15.1% in men and 7.1% in women), indicating a lower prevalence of BD than that reported for the Japanese population, which was of 29.3%, being higher for men than for women (40.6% vs. 18.7%, respectively). The measurement of excessive consumption followed the same CDC criteria and included the last 30 days; however, for the sake of convenience, the study was carried out online in a population over 18 years of age, to which only one-fifth of those invited to participate responded [65]. The prevalence of excessive alcohol consumption varies according to the way in which consumption is measured. The prevalence of excessive consumption changed in 2020. It increased in some people and decreased in others, due to the social circumstances of isolation related with the pandemic [67,68].

This is one of the first studies to identify the association between physical violence perpetrated by an intimate partner and BD according to sex in the Peruvian population. However, despite being a novel and nationally-representative study, it has some limitations. One of them is the design of the study which, being cross-sectional, did not allow to show the direction in which the variables interact, the path between BD and victimization and vice versa, and even the exercise of bidirectional violence within the couple. Additionally, the data used to conduct the study did not specify the gender of the interviewees’ partner or their sexual orientation. It is known that there are homosexual and heterosexual couples in Peru, so it is important to be able to discriminate between violence from both types of couples, information that is not available for this study. Currently, an increase in same-sex couples’ violence has been observed, but there are still limitations for the study of these cases [53]. During the investigation, we could detect another important limitation, which is the lack of questions about the emotional or sexual violence exerted on men, which restricts the investigation only to physical violence and limits the real impact of all types of violence against men. This aspect is relevant, since a systemic review found that the most prevalent violence exerted on men is psychological [64]. Additionally, there is a gap in information about the identity of the perpetrator of violence. Even though questions inquire if, in the last 12 months, they have been living with a partner, and if they have been a victim of physical violence by them, it does not specify the time period in which the physical violence occurred, if there has been more than one partner in that period of time, nor if it was their current partner but also a previous partner [60]. On the other hand, neither religion nor the consumption of other substances were included in the variables.

Despite the training and standardization of interviewers and supervisors of the Peruvian Demographic Health Survey in highly sensitive issues such as violence, the validation and review of how questions were asked, the selection of language, the training in situations such as the presence of other people during the interview, and the interviewer’s attitude, can influence the estimation of violence, possibly biased toward violence underestimation. This also affects in more recent and severe cases [69,70].

A memory bias could have been present due to the period of time considered in the questions, as well as a desirability bias, since the interviewee could have thought that some of the answers were more socially acceptable. Due to the fact that the information had already been collected, no further investigation could be carried out on aspects not included in the original study (such as the number of partners, whether the aggressor partner lived with the victim, or if the aggressor’s partner and the victim were married or cohabiting, in addition to other sociodemographic information about the couple). Likewise, no information could be gathered about participants’ history of violence during childhood, since the questions about violence were limited to the last 12 months.

## 5. Recommendations

The development of longitudinal studies including early exposure to violence, a history of post-traumatic stress disorder, depression, and patterns of alcohol consumption that delve into the causal association between BD and physical aggression, is suggested [35]. Additionally, the development of comparative studies between the years before and after the pandemic is recommended. Gender-based strategies concerning health should be applied to improve visibility and prevent intimate partner aggression against both men and women. Early identification of intimate partner violence and strategies to prevent BD in victims of violence should be our target.

## 6. Conclusions

The results of this study found an association between having been a victim of physical violence by an intimate partner and BD, in both men and women, in the Peruvian population over 15 years of age. It was identified that the prevalence of physical violence victimization was higher in women than in men (4.7% vs. 3%, respectively). Male and female victims of aggression were more susceptible to present an excessive consumption of alcohol, which was higher in men than in women. Further investigation on these findings is needed, with longitudinal studies taking into account genetic characteristics, as well as social and cultural determinants in various populations.

## Figures and Tables

**Figure 1 ijerph-19-14403-f001:**
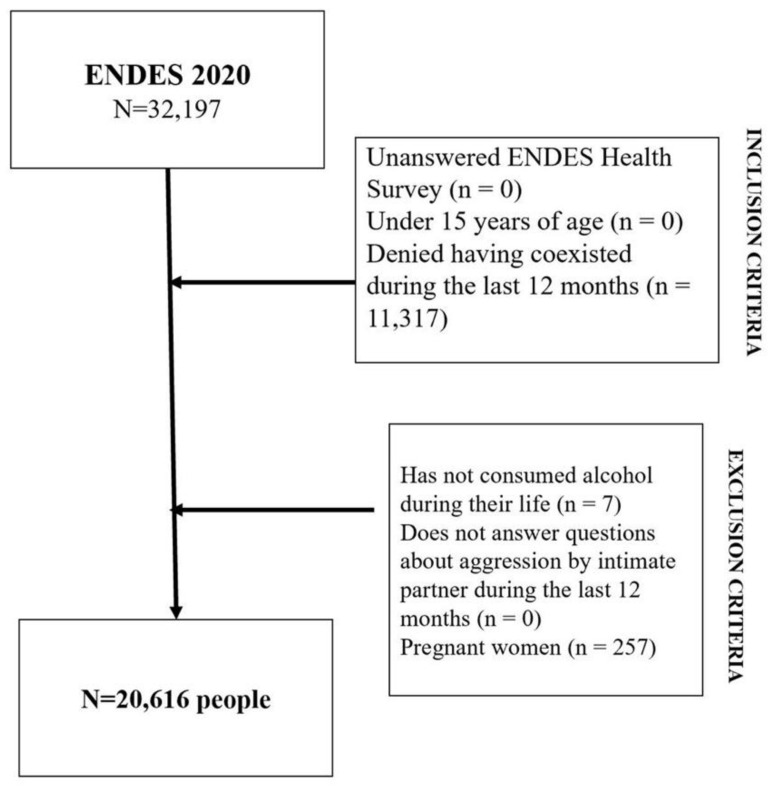
Flowchart showing data obtained from the population through inclusion and exclusion criteria.

**Table 1 ijerph-19-14403-t001:** General characteristics of the population older than 15 years of age in Peru 2020 (*n* = 20,616).

		*n*	*%* a	CI (95%)	
				LL	UL
**Age in years**					
	Between 15 and 24	1593	8.4	7.8	9.1
	Between 25 and 34	6148	24.8	23.8	25.7
	Between 35 and 49	7190	34.9	33.8	35.9
	Older than 50	5685	31.9	30.8	33.1
**Sex**					
	Male	9923	50.7	49.6	51.7
	Female	10,693	49.3	48.3	50.4
**Ethnic self-identification**					
	Quechua	6452	23.9	4.7	5.7
	Aymara	765	1.9	22.9	25.0
	Native or indigenous	324	0.8	1.6	2.2
	Black/brown/zambo/mulatto	2097	11.1	0.6	1.0
	Caucasian	1163	6.9	10.5	11.8
	Mestizo	8477	48.4	6.3	7.5
	Part of one town or others	347	1.8	47.2	49.6
	Does not know/does not answer	991	5.2	1.6	2.2
**Socioeconomic status**					
	Bottom Quintile	6988	19.6	18.9	20.4
	Second Quintile	5158	20.2	19.2	21.2
	Third Quintile	3783	21.2	20.2	22.2
	Fourth Quintile	2695	19.6	18.6	20.7
	Top Quintile	1992	19.3	18.2	20.5
**Residence**					
	Urban area	12,786	78.9	78.2	79.6
	Rural area	7830	21.1	20.4	21.8
**Natural region**					
	Coast	7717	61.8	60.6	62.9
	Mountains	7786	25.1	24.0	26.3
	Jungle	5113	13.1	12.4	13.9
**Marital status**					
	Single	98	0.8	0.6	1.1
	Married or living together	19,823	96.0	95.5	95.5
	Divorced, widowed, or separated	695	3.2	2.9	3.6
**Educational level**					
	Pre-school or less	743	2.8	2.5	3.1
	Primary education	5479	21.2	20.4	22.1
	Secondary education	8920	43.7	42.6	44.8
	Non-university higher education	3030	17.2	16.3	18.2
	University education or more	2444	15.1	14.2	16.0

a All percentages are weighted. CI = Confidence interval, UL = Upper limit, LL = Lower limit.

**Table 2 ijerph-19-14403-t002:** Problem behaviors according to sex in the population older than 15 years of age in Peru 2020 (*n* = 20,616).

	Female *n* = 10,693 (49.4%)				Male *n* = 9923 (50.7%)				
	*n*	*%* a	CI 95%		*n*	*%* a	CI (95%)		*p* c
			LL	UL			LL	UL	
**Type of alcoholic beverage consumed in the last 30 days d**									
Beer	757	7.6	6.7	8.5	1318	14.5	13.2	15.9	
Wine/*cachina*/champagne	412	5.2	4.4	6.1	220	3.0	2.4	3.7	
Fermented *chicha de jora* or fermented *masato*	67	0.7	0.5	1.0	81	0.7	0.5	1.0	
*Yonque*/cane spirit	33	0.2	0.1	0.2	166	1.0	0.8	1.3	
Anisette/Whiskey/Pisco/Rum/Other	126	1.9	1.4	2.5	184	2.8	2.2	3.5	
None	9298	84.6	83.0	86.0	7954	78.0	76.2	79.7	<0.001
**Presence of depressive symptoms in the last 14 days e**									
Moderate symptoms or more	775	7.5	6.7	8.3	364	3.9	3.4	4.6	
No symptoms or minimal symptoms	9918	92.5	91.7	93.3	9559	96.1	95.4	96.6	<0.001
**Tobacco consumption during the last 30 days**									
Yes	23	0.2	0.1	0.4	180	2.1	1.6	2.6	
No	10,670	99.8	99.6	99.9	9743	97.9	97.4	98.4	<0.001
**Physical violence by the intimate partner in the last 12 months**									
Yes	529	4.7	4.1	5.4	238	3.0	2.4	3.7	
No	10,164	95.3	94.7	95.9	9685	97.0	96.3	97.6	<0.001

CI = Confidence interval, UL = Upper limit, LL= Lower limit. a All percentages are weighted. c Chi2 analysis with Rao-Scott correction. d The beverage most frequently consumed in the last 30 days. e Based on the PHQ9 scale cut-off point 10 or more.

**Table 3 ijerph-19-14403-t003:** Association between sociodemographic characteristics and excessive alcohol consumption in individuals (*n* = 20,616).

	Excessive Alcohol Consumption												
	Yes				No								
	*n* = 2104 (10.2%)				*n* = 18,512 (89.8%)								
			95% CI				95% CI			PRc	95% CI		*P* m
	*n*	% b	LL	UL	*n*	% b	LL	UL	*p* h		LL	UL	
**Sex**													
Female	687	7.1	6.2	9.0	10,006	92.9	92.0	93.8		Ref			
Male	1417	15.1	13.8	16.5	8506	84.9	83.5	86.3	<0.001	2.1	1.9	2.4	<0.001
**Age**													
Older than 50 years of age	362	6.9	5.8	8.2	5323	93.1	91.8	94.2		Ref.			
Between 35 and 49 years of age	758	10.9	9.6	12.4	6432	89.1	87.6	90.4	<0.001	0.8	0.6	1.4	0.049
Between 25 and 34 years of age	803	15.9	14.2	17.6	5345	84.1	82.4	85.8		1.1	0.9	1.4	0.339
Between 15 and 24 years of age	181	14.1	11.3	17.6	1412	85.9	82.4	88.8		2.0	1.6	2.7	<0.001
**Self-identification**													
Quechua	538	9.3	8.1	10.8	5914	90.7	89.2	91.9		Ref			
Aymara	61	8.8	5.9	12.9	704	91.2	87.1	94.1		0.9	0.6	1.4	0.755
Native or indigenous	51	12.5	7.9	19.2	273	87.5	80.8	92.1		1.3	0.8	2.1	0.225
Black/brown/zambo/mulatto	224	13.3	10.9	16.1	1873	86.7	83.9	89.1		1.4	1.1	1.8	0.004
Caucasian	116	12.1	9.3	15.6	1047	87.9	84.4	90.7		1.3	1.0	1.73	0.083
Mestizo	990	11.7	10.5	13.0	7487	88.3	87.0	89.5		1.3	1.1	1.5	0.007
Part of one town or another	49	17.1	10.6	26.4	298	82.9	73.6	89.4		1.8	1.1	2.9	0.014
Does not know/does not answer	75	6.9	5.0	9.5	916	93.1	90.5	95.0	0.001	0.7	0.5	1.1	0.092
**Socioeconomic status**													
Bottom Quintile	452	6.3	5.4	7.4	6536	93.7	92.6	94.6		Ref.			
Second Quintile	564	10.4	9.2	11.9	4594	89.6	88.1	90.8		1.6	1.4	2.0	<0.001
Third Quintile	436	11.9	10.3	13.8	3347	88.1	86.3	89.7		1.9	1.5	2.3	<0.001
Fourth Quintile	368	12.6	10.7	14.8	2327	87.4	85.2	89.3		2.0	1.6	2.5	<0.001
Top Quintile	284	14.4	11.9	17.2	1708	85.6	82.8	88.1	<0.001	2.3	1.8	2.9	<0.001
**Residence**													
Rural area	590	7.7	6.6	9.0	7240	92.3	91.0	93.4		Ref.			
Urban area	1514	12.1	11.0	13.3	11,272	87.9	86.8	89.0	<0.001	1.6	1.3	1.9	<0.001
**Natural region**													
Coast	939	12.3	11.0	13.8	6778	87.7	86.2	89.0		Ref.			
Mountains	568	8.0	6.9	9.2	7218	92.0	90.8	93.1		0.6	1	0.8	<0.001
Jungle	597	11.7	10.1	13.5	4516	88.3	86.5	90.0	<0.001	0.9	1	1.1	0.572
**Educational level**													
Preschool or less	23	3.3	1.9	5.6	720	96.7	94.4	98.1		Ref.			
Primary education	321	6.2	5.2	7.3	5158	93.8	92.7	94.8		1.9	1.1	3.3	0.025
Secondary education	1043	13.3	12.0	14.7	7877	86.7	85.3	88.1		4.1	2.4	7.1	<0.001
Non-university higher education	392	11.5	9.8	13.5	2638	88.5	86.5	90.2		3.5	2.0	6.2	<0.001
University education or more	325	13.0	10.7	15.7	2119	87.0	84.3	89.3	<0.001	4.0	2.3	7.1	<0.001
**Cigarette consumption in the last 30 days**													
Does not consume	133	10.9	10.0	11.8	18,379	89.1	88.2	90.0		Ref.			
Consumes	133	34.8	25.4	45.5	133	65.2	54.5	75.0	<0.001	3.2	2.4	4.3	<0.001
**Presence of depressive symptoms in the last 14 days r**													
No symptoms or minimal symptoms	2012	11.3	10.4	12.3	17465	88.7	87.7	89.7		Ref.			
Moderate symptoms or more	92	8.9	6.6	11.9	1047	91.1	88.1	93.4	0.113	0.8	0.6	1.1	0.118
**Type of beverage consumed x**													
Beer	1415	65.5	62.2	68.7	660	34.5	31.3	37.8					
Wine/*cachina*/champagne	278	41.5	35.4	47.9	354	58.5	52.1	64.6					
Fermented *chicha de jora* or fermented *masato*	99	66.7	56.2	75.8	49	33.3	24.2	43.9					
*Yonque*/cane spirit	119	65.0	55.5	73.4	80	35.0	26.6	44.5					
Anisette/Whiskey/Pisco/Rum/Other	193	58.1	49.9	65.8	117	41.9	34.2	50.1	<0.001				

h Pearson Chi2 analysis with Rao–Scott correction. b All percentages are weighted. x Only people who have consumed alcoholic beverages in the last 30 days are considered. r Obtained from PHQ-9 scale. m PRc: prevalence ratio.

**Table 4 ijerph-19-14403-t004:** Association between intimate partner physical aggression and excessive alcohol consumption in the population over 15 years of age in Peru 2020 (*n* = 20,616).

	General Model								Model Based on Sex															
									Male								Female							
	PRc o	95% CI		*p*	PRa m	95% CI		*p*	PRc o	95% CI		*p*	PRa n	95% CI		*p*	PRc o	95% CI		*p*	PRa n	95% CI		*p*
		LL	UL			LL	UL			LL	UL			LL	UL			LL	UL			LL	UL	
**Intimate partner aggression during the last 12 months**																								
No	Ref								Ref								Ref				Ref			
Yes	1.9	1.5	2.5	<0.001	1.9	1.4	2.5	<0.001	2.2	1.6	3.2	<0.001	1.9	1.3	2.9	0.001	1.9	1.3	2.7	0.001	1.8	1.1	2.8	0.013

PRc; Prevalence ratio in crude model. PRa; Prevalence ratio in adjusted model. m Generalized linear model of the link-log Poisson family, adjusted for sex, age, self-identification, socioeconomic status, educational level, residence, presence of depressive symptoms in the last 14 days, and tobacco consumption in the last 30 days. n Generalized linear model of the link-log Poisson family, adjusted for age, self-identification, socioeconomic status, educational level, residence, presence of depressive symptoms in the last 14 days, and tobacco consumption in the last 30 days. o Generalized linear model of the link-log Poisson family crude model. CI: Confidence Interval, LL: Lower limit, UL: Upper Limit.

## Data Availability

The data presented in this study are openly available at http://iinei.inei.gob.pe/microdatos/ (accessed on 9 October 2022).

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
