# Peer review of "Is There an Association between Being a Victim of Physical Violence by Intimate Partner and Binge Drinking in Men and Women? Secondary Analysis of a National Study, Peru 2020"

_ijerph, 2022, doi:10.3390/ijerph192114403_

Round 1
Reviewer 1 Report
This study considers an important and under studied phenomenon: the relationship between intimate partner violence and binge drinking. The authors have a unique data set with which to conduct the analysis. Their argument will be much stronger if they address two key issues/limitations:
1. First, they do note the problem of the directional nature of the phenomena (e.g. does excessive drinking trigger domestic violence or is it a response to it)...I think they need to unpack this further in both the introduction and the discussion, especially given the correlation between drinking and perpetrating DV and also the correlation between drinking as a way to self-medicate by victim/survivors. I think their finding is still sound, but they need to make more clear the empirical relationship between these 2 variables.
2. Second, the authors' argument is open to critique given that accept the data without considering the differences between situational couple violence and intimate terrorism, which are often obscured by the kinds of measures utilized in this study. In order to address this critique, I recommend that the authors address this tension and critiques of these types of measures and offer this as a possible limitation that needs to be considered in the analysis.
Author Response
Response to Reviewer 1 Comments
Reviewer 1
This study considers an important and under studied phenomenon: the relationship between intimate partner violence and binge drinking. The authors have a unique data set with which to conduct the analysis. Their argument will be much stronger if they address two key issues/limitations:
- First, they do note the problem of the directional nature of the phenomena (e.g. does excessive drinking trigger domestic violence or is it a response to it)...I think they need to unpack this further in both the introduction and the discussion, especially given the correlation between drinking and perpetrating DV and also the correlation between drinking as a way to self-medicate by victim/survivors. I think their finding is still sound, but they need to make more clear the empirical relationship between these 2 variables.
Thank you for your valuable suggestions we included the bidirectionality and a cyclical temporal dynamic between excessive alcohol consumption and intimate partner violence suggestion in the second to last paragraph in the introduction.
“Evidence suggests a bidirectionality and cyclical temporal dynamic between BD and IPV. The mechanisms in BD to victimization include settings where victims and perpetrators drink heavily, which increases disinhibited behaviours, as well as increased vulnerability to victimization. Contradictory IPV victims who commit BD after the aggression seek for an avoidant coping strategy, a distraction of psychological symptoms. Research in this last direction is less explored and there are some discrepant results specially as these studies are been done predominantly in women (1)
We included also in the discussion the following paragraph in relation of the suggestion:
“Other studies comment upon the incidence of binge drinking after being an IPV victim. It is known that cycle violence and cycle drinking are common conducts of an abusive lifestyle, however, its noted that being a victim for IPV is a risk factor for a greater alcohol consumption. IPV suggests more exposure to general violence a highly stressful situations, alluding to a cumulative stress-drinking as a maladaptive copying mechanism. The stress drinking can be delivered by victims as well as the perpetrator, both with purpose of eliminating the memory of the previously committed violence. Also, alcohol is not exclusive to bingeing, food and substance abuse are included as well.(2, 3) On the other hand, intimate terrorism as well as negative social interactions such as violence at bars and parties for the women population may contribute to the abusive drinking behaviour (4).”
-
- Second, the authors' argument is open to critique given that accept the data without considering the differences between situational couple violence and intimate terrorism, which are often obscured by the kinds of measures utilized in this study. In order to address this critique, I recommend that the authors address this tension and critiques of these types of measures and offer this as a possible limitation that needs to be considered in the analysis.
Thank you very much for your comment, in the limitation section we included the gap of information for further research about the identity of the perpetrator in partner violence, including intimate terrorism : not only their current partner but also their previous partners.
“Also there is a gap of information about the identity of the perpetrator of violence. Eventhough questions inquire if in the last 12 months they have been living with a partner, and if they have been a victim of physical violence by them, it does not specify the time period in which the physical violence occurred, if there has been more than one partner in that period of time nor if it was their current partner but also a previous partner(4). “
1 Dardis CM, Ullman SE, Rodriguez LM, Waterman EA, Dworkin ER, Edwards KM. Bidirectional associations between alcohol use and intimate partner violence and sexual assault victimization among college women. Addict Behav. 2021;116:106833.
- Yalch MM, Christodoulou J, Rotheram-Borus MJ, Tomlinson M. Longitudinal Association Between Intimate Partner Violence and Alcohol Use in a Population Cohort of South African Women. Journal of Interpersonal Violence.
- Ullman SE, Sigurvinsdottir R. Intimate Partner Violence and Drinking Among Victims of Adult Sexual Assault. J Aggress Maltreat Trauma. 2015;24(2):117-30.
- Johnson MP, Leone JM, Xu YL. Intimate Terrorism and Situational Couple Violence in General Surveys: Ex-Spouses Required. Violence against Women. 2014;20(2):186-207.
- Second, the authors' argument is open to critique given that accept the data without considering the differences between situational couple violence and intimate terrorism, which are often obscured by the kinds of measures utilized in this study. In order to address this critique, I recommend that the authors address this tension and critiques of these types of measures and offer this as a possible limitation that needs to be considered in the analysis.
-

Reviewer 2 Report
The paper investigates the relationship between being a victim of physical violence by an intimate partner and Binge Drinking in men and women in Peru in 2020. Despite it has several limitations, I want to thank the Authors, because they analyzed a very peculiar aspect of violence, including men's side.
My suggestions are below:
- pag.3, line of the reference #37: a full stop is missing.
- Please, clarify the reason why you excluded pregnant women.
- Please, clarify why you decided to consider only the last 12 months, and only the last 30 days for alcohol consumption.
- I think that interviews face-to-face are a limitation: it could have been a bias, because interviews were not anonymous. Please, could you explain better this aspect?
- How was taken the consent? Especially by 15-18-year-old sample?
- tab 3: make explicit LL - UL in the legend, please.
For this reason, I am asking to improve the description of methods and results, better specifying the reason of some choices taken.
Thank you for your contribution.
Best regards.
Author Response
Answer to comments of Reviewer 2 .
Reviewer 2
The paper investigates the relationship between being a victim of physical violence by an intimate partner and Binge Drinking in men and women in Peru in 2020. Despite it has several limitations, I want to thank the Authors, because they analyzed a very peculiar aspect of violence, including men's side.
My suggestions are below:
1.pag.3, line of the reference #37: a full stop is missing.
Your suggestion was included. In the draft version of the manuscript attached to this answer.
- Please, clarify the reason why you excluded pregnant women
The possibility of a biased response regarding alcohol consumption during pregnancy is high because of stigmatization and underreport, as well as the variation of standarized measurements for binge drinking throughout the pregnancy(5, 6).
3.Please, clarify why you decided to consider only the last 12 months in violence, and only the last 30 days for alcohol consumption.
When it comes to the timeline of alcohol consumption as binge drinking and intimate partner physical violence, it is a cycle, not a linear timeline. With this said, regarding the time of when the aggression is commented (being the last twelve months) it is suitable to consider an association with a 30 day mark in binge drinking. Establishing association is not limited to a one directional relation.(7) Also our study is a secondary analysis of the Demographic Health Survey , and health indicators of violence in men and women are measure according to DHS parameters (in the last 12 months while they live together with a partner). https://dhsprogram.com/Data/Guide-to-DHS-Statistics/index.cfm, and binge drinking is a standardized measure (if they have use alcohol in the last 30 days on more than one occasion, report of how much and what type of liquor they consumed on the occasion where they drank the most).
- I think that interviews face-to-face are a limitation: it could have been a bias, because interviews were not anonymous. Please, could you explain better this aspect?
Thank you very much for your comment, we will include in the limitation section this important issue. Despite the training and standardization of interviewers and supervisors of the Peruvian Demographic Health Survey in highly sensitive issues such as violence, the validation and review of how questions were asked, the selection of language, the training in situations such as the presence of other people during the interview; the interviewer's attitude can influence the estimation of violence, possibly biased toward violence underestimation. This also affects in more recent and severe cases(8, 9)
- How was taken the consent? Especially by 15-18-year-old sample?
As it is stated in the Peruvian DHS obtained informed consent and assent from each participant and anonymized the data set (http://iinei.inei.gob.pe/microdatos/). Moreover, the data for this study was obtained from the DHS freely access data (http://iinei.inei.gob.pe/microdatos/). Since we used secondary data for analysis and the research protocol was reviewed and approved with the by the ethical committee of the Universidad Peruana de Ciencias Aplicadas. FCS-SCEI/1153-11-21
- tab 3: make explicit LL - UL in the legend, please.
Your suggestion was included In the draft version of the manuscript attached to this answer
For this reason, I am asking to improve the description of methods and results, better specifying the reason of some choices taken.
5. Lee SH, Shin SJ, Won SD, Kim EJ, Oh DY. Alcohol Use during Pregnancy and Related Risk Factors in Korea. Psychiatry Investigation. 2010;7(2):86-92.
6. Iversen ML, Sorensen NO, Broberg L, Damm P, Hedegaard M, Tabor A, et al. Alcohol consumption and binge drinking in early pregnancy. A cross-sectional study with data from the Copenhagen Pregnancy Cohort. Bmc Pregnancy and Childbirth. 2015;15.
7.Wilson IM, Graham K, Taft A. Living the cycle of drinking and violence: A qualitative study of women's experience of alcohol‐related intimate partner violence. Drug and alcohol review. 2017;36(1):115-24.
8. Abramsky T, Harvey S, Mosha N, Mtolela G, Gibbs A, Mshana G, et al. Longitudinal inconsistencies in women's self-reports of lifetime experience of physical and sexual IPV: evidence from the MAISHA trial and follow-on study in North-western Tanzania. Bmc Womens Health. 2022;22(1).
9. Singh A, Kumar K, Arnold F. How Interviewers Affect Responses to Sensitive Questions on the Justification for Wife Beating, the Refusal to have Conjugal Sex, and Domestic Violence in India. Stud Fam Plann. 2022;53(2):259-79.A draft vertion of the manuscript with included suggestion is attach to this answer

Round 2
Reviewer 1 Report
I appreciate the revisions the authors provided. However, they still do not adequately address the critique that these kinds of scales skew toward measuring situational couple violence. Additionally, though they have added this sentence: The stress drinking can be delivered by victims as well as the perpetrator, both with purpose of eliminating the memory of the previously committed violence. This is not helpful, it's actually quite concerning. Though yes, victims may drink to block the memory of the abuse, perpetrators often encourage binge drinking in their victims in order to facilitate the abuse and reduce the victim's ability to remember exactly what happened, making them vulnerable to gas lighting and reducing the likelihood that they would report the abuse to anyone. Given that these kinds of statements can easily be misinterpreted, and the authors are open to serious critique, I'd suggest they consider more thoroughly than just a few sentences how to address these potential critiques throughout the paper.
Author Response
Reviewer 1
- I appreciate the revisions the authors provided. However, they still do not adequately address the critique that these kinds of scales skew toward measuring situational couple violence. Additionally, though they have added this sentence: The stress drinking can be delivered by victims as well as the perpetrator, both with purpose of eliminating the memory of the previously committed violence. This is not helpful, it's actually quite concerning. Though yes, victims may drink to block the memory of the abuse, perpetrators often encourage binge drinking in their victims in order to facilitate the abuse and reduce the victim's ability to remember exactly what happened, making them vulnerable to gas lighting and reducing the likelihood that they would report the abuse to anyone. Given that these kinds of statements can easily be misinterpreted, and the authors are open to serious critique, I'd suggest they consider more thoroughly than just a few sentences how to address these potential critiques throughout the paper.
Answer to comments and suggestions of reviewer 1
- We appreciate your valuable suggestion, which has been enlightening. We have deepened the information about the proposed mechanisms to explain the relationship between being a victim of intimate partner violence and excessive alcohol consumption considering the interaction between different levels of intertwined individual, relationship, cultural and social factors and we have considered the bidirectionality and the cycle of violence and the continuity of alcohol consumption. It is included in the third and fourth paragraphs of the discussion section.
“Some proposed mechanisms for understanding BD as a consequence of partner or ex-partner victimization in men or women include the interaction of factors at different levels, such as individual characteristics and background (e.g., genetics, personality, level of self-esteem, adverse childhood events, and alcohol consumption) that may condition a greater predisposition to conscious or unconscious binge drinking in response to negative psychological symptoms resulting from victimization, either from the incident or from incident-related problems, according to the dose-response hypothesis(1, 2). In addition to allowing that in some cases this excessive consumption, which was given as an action to buffer the symptoms, instead of achieving the desired effect results in more negative emotions generating a vicious cycle (3).
This interaction encompasses also factors at the relationship level, such as type of partner (e.g., current partner, ex-partner, same sex partner), alcohol consumption (habitual use, frequency of BD, relation to the perpetration of violence), characteristics of the place where the violence and excessive alcohol consumption occur, whether in private and alone or in bars, parties or social gatherings or others (4), as well as social factors and cultural norms of each setting such as social stigmatization and the resources available to identify, prevent and manage IPV and BD situations, which may vary indifferent populations and cultures(5).”
The manuscript with the included paragraph is attached to this answer.
